# A Leaky-Wave Analysis of Resonant Bessel-Beam Launchers: Design Criteria, Practical Examples, and Potential Applicationsat Microwave and Millimeter-Wave Frequencies

**DOI:** 10.3390/mi13122230

**Published:** 2022-12-15

**Authors:** Edoardo Negri, Walter Fuscaldo, Paolo Burghignoli, Alessandro Galli

**Affiliations:** 1Department of Information Engineering, Electronics and Telecommunications, Sapienza University of Rome, 00184 Rome, Italy; 2Istituto per la Microelettronica e Microsistemi, Consiglio Nazionale delle Ricerche, 00133 Rome, Italy

**Keywords:** Bessel beams, leaky waves, wireless power transfer, metasurfaces

## Abstract

Resonant Bessel-beam launchers are low-cost, planar, miniaturized devices capable of focusing electromagnetic radiation in a very efficient way in various frequency ranges, with recent increasing interest for microwave and millimeter-wave applications (i.e., 3–300 GHz). In recent years, various kinds of launchers have appeared, with different feeding mechanisms (e.g., coaxial probes, resonant slots, or loop antennas), field polarization (radial, azimuthal, and longitudinal), and manufacturing technology (axicon lenses, radial waveguides, or diffraction gratings). In this paper, we review the various features of these launchers both from a general electromagnetic background and a more specific *leaky-wave* interpretation. The latter allows for deriving a useful set of *design rules* that we here show to be applicable to any type of launcher, regardless its specific realization. Practical examples are discussed, showing a typical application of the proposed design workflow, along with a possible use of the launchers in a modern context, such as that of wireless power transfer at 90 GHz.

## 1. Introduction

Any kind of wave (light, sound, etc.), initially confined to a finite area on a transverse plane, will be subject to diffractive spreading as it propagates outward from the source to free space. As a result, diffraction effects limit all applications in which it is required to maintain a spatial transverse confinement of a *beam* (i.e., a monochromatic electromagnetic wave with a well defined propagation axis) over a considerable distance. Such applications comprise but are not limited to wireless power transfer, free-space communications, medical imaging, radiometry, etc. For this reason, many efforts were made to find wave solutions that do not undergo diffraction.

*Localized waves* or *nondiffracting waves* are the largest family of *diffraction-free* solutions to the wave equation. As the name suggests, these waves ideally do not diffract, thus they maintain their transverse profile for an infinite distance from the radiating aperture. In principle, this remains true only for an infinite-size radiating aperture. Conversely, for a truncated radiating aperture, the localized waves can resist diffraction only up to a finite distance [1,2].

*Bessel beams* are one of the most known examples of localized waves. Bessel beams are monochromatic solutions of the Helmholtz equation in a cylindrical reference frame with very interesting focusing and limited-diffraction properties. These features attracted much interest for different applications in optics during the 1980s [3]. Currently, Bessel beams are studied and analyzed in the microwave and millimeter-wave ranges due to their usage in near-field focusing applications [4]. In these frequency ranges, Bessel beams are typically generated through planar structures, also known as *Bessel-beam launchers*. Such launchers are further distinguished between the *resonant* kind and the *wideband* kind. The former are realized through compact planar devices with only a few wavelengths of lateral size; they typically feature a narrow-band behavior due to their resonant character [5,6]. The latter, instead, feature a wider bandwidth at the expense of a lateral size that can be as large as several wavelengths [7,8,9]. Both kinds have been experimentally validated [6,7,10], but here we will deal only with the resonant type. Depending on the application, such as in WPT links for wearable devices, resonant Bessel-beam launchers are preferred to their wideband counterpart thanks to their compact implementation. Due to the lack of a comprehensive analysis of such devices in the available literature, only resonant Bessel-beam launchers are analyzed in this work (the interested reader can find information about wideband Bessel-beam launchers in, e.g., [4]).

Resonant Bessel-beam launchers can be realized in different ways, although they all exhibit azimuthal symmetry and share the same architecture: a circular metallic cavity excited by a simple dipole-like source at the center, whose upper plate is replaced by a partially reflecting sheet (PRS). The cavity height *h* plays a key role. When the cavity height is deeply subwavelength, h≪λ, Bessel-beam launchers require a capacitive PRS and can only work with the *fundamental*, zeroth-order transverse-magnetic (TM) (with respect to the vertical *z*-axis) leaky mode [11]. When the cavity height is half a wavelength, h≃λ/2, the PRS can be either inductive or capacitive [11], and the launchers can work with either higher-order transverse-electric (TE) or TM leaky modes. Because ohmic losses in waveguide-like devices as Bessel-beam launchers decrease with the cavity height [12], higher-order resonant Bessel-beam launchers are usually preferred at millimeter-wave or sub-millimeter wave frequencies, where dissipative effects in metals and dielectrics may become important. In this paper, we will thus focus on such kinds of launchers.

Higher-order resonant Bessel-beam launchers can further be divided in two different kinds according to their feeding mechanism. In particular, if a vertical magnetic or electric dipole as source is used, then a TE- [13,14] or TM-polarized [6] Bessel beam is generated.

In the available literature, resonant Bessel-beam launchers in the microwave and millimeter-wave frequency range (i.e., 3–300 GHz) have been widely analyzed as stand-alone devices [5,6,13,15] and as elements in a WPT link [16,17], but a comprehensive analysis of this kind of launchers in terms of polarization, design, and physical implementations is still lacking. The main purpose of this work is indeed to provide a simple, yet rigorous step-by-step design workflow for differently polarized resonant Bessel-beam launchers based on leaky-wave theory. Theoretical results are exploited to derive design guidelines that are here applied for the physical implementation of an original device operating at 90 GHz. The radiating performance of the device is then corroborated through a comparison between full-wave and numerical results based on leaky-wave theory. Finally, different applications for Bessel beams are presented. In particular, the performance of a WPT link at 90 GHz realized through the resonant Bessel-beam launchers designed in this work is shown.

The paper is organized as follows. In Section 2, the theoretical framework underlying the physics and working principle of Bessel beams is first described. In Section 3, focused radiation from resonant Bessel-beam launchers is explained under a leaky-wave interpretation. This powerful analytical model allows for deriving simple design criteria to realize launchers with certain desired focusing properties. In Section 4, it is shown how the general framework developed in Section 2 and Section 3 can fruitfully be applied to design practical devices that can easily be microfabricated through printed-circuit board (PCB) and/or standard photolithographic techniques, depending on the targeted frequency range. In Section 5, a brief discussion is provided about the possible application scenarios of the proposed devices. Finally, conclusions are drawn in Section 6.

## 2. Theoretical Framework

We follow [1] and start from the homogeneous wave equation in a cylindrical reference frame (ρ,ϕ,z) under the hypothesis of axial symmetry (∂/∂ϕ=0):(1)$$\left(\frac{\partial^{2}}{\partial \rho^{2}}+\frac{1}{\rho }\frac{\partial }{\partial \rho }+\frac{\partial^{2}}{\partial z^{2}}-\frac{1}{c_{0}^{2}}\frac{\partial^{2}}{\partial t^{2}}\right)\psi \left(\rho ,z,t\right)=0$$
where $c_{0}$ is the speed of light in vacuum and ψ is a component of the electromagnetic field or potential. In order to find a general solution to (Equation 1), a spectral method is applied: a Hankel transform with respect to ρ, and two Fourier transforms with respect to *z* and *t* lead to the following integral expression (a time dependence $e^{j\omega t}$ of the potential is assumed and suppressed throughout the paper):(2)$$\psi \left(\rho ,z,t\right)=\int_{-\infty }^{+\infty }\int_{-\infty }^{+\infty }\int_{-\infty }^{+\infty }k_{\rho }J_{0}\left(k_{\rho }\rho \right)e^{-jk_{z}z}e^{j\omega t}\tilde{\psi }\left(k_{\rho },k_{z},\omega \right)dk_{\rho }dk_{z}d\omega$$
where $k_{\rho }$ and $k_{z}$ are the radial and longitudinal wavenumbers, respectively, ω is the angular frequency, and $\tilde{\psi }\left(k_{\rho },k_{z},\omega \right)$ is the generalized transform of ψ(ρ,z,t). One order of integration can be skipped by exploiting the separation relation $k_{0}^{2}=k_{\rho }^{2}+k_{z}^{2}$, where $k_{0}=\omega /c_{0}$ is the vacuum wavenumber. The integral in (Equation 3) can then be expressed in terms of an arbitrary spectral function $S(k_{\rho },\omega )$ as:(3)$$\psi \left(\rho ,z,t\right)=\int_{0}^{+\infty }\int_{-k_{0}}^{+k_{0}}k_{\rho }J_{0}\left(k_{\rho }\rho \right)e^{-jk_{z}z}e^{j\omega t}S\left(k_{\rho },\omega \right)dk_{\rho }d\omega$$
where we have limited the domain of ω and $k_{\rho }$ to consider only positive frequencies $\omega \in R^{+}$ and fast, radiating waves $k_{\rho }\leq k_{0}$. The wave solution in (Equation 3) is rather general and expresses any solution of the Helmholtz equation in an axially symmetric reference frame, as a superposition of radiating plane waves. In order to obtain a localized solution, a linear coupling between ω and $k_{\rho }$ has to be imposed [18]. This constraint is immediately obtained by enforcing the propagation-invariant property (which distinguishes a *localized* solution from any wave solution) to the wave solution $\psi \left(\rho ,z,t\right)=\psi \left(\rho ,t-z/v_{z}\right)$, where $v_{z}$ is a constant (which implies rigid transport) representing the wave velocity along the *z*-axis.

Among localized waves, Bessel beams are particularly interesting solutions due to their self-healing property [3,19,20]. They are obtained from a spectrum equal to:(4)$$S\left(k_{\rho },\omega \right)=\frac{\delta (k_{\rho }-k_{0}\sin\theta_{0})}{k_{\rho }}\delta \left(\omega -\omega_{0}\right).$$
where δ(·) is the Dirac delta function. Equation (Equation 4) clearly implies that a Bessel beam is a monochromatic solution in $\omega_{0}=2\pi f_{0}$ ($f_{0}$ being the working frequency) with $k_{\rho }=k_{0}\sin\theta_{0}=\left(\omega_{0}/c_{0}\right)\sin\theta_{0}$, where $0<\theta_{0}<\pi /2$ is the so-called *axicon angle*, defined with respect to the vertical *z*-axis. Therefore, a Bessel beam can be represented as a superposition of plane waves with wavenumbers lying on the surface of a cone with aperture angle 2$\theta_{0}$. From the separation relation, we also have $k_{z}=k_{0}\cos\theta_{0}$, thus a linear coupling between ω and $k_{z}$ is required for obtaining localized-wave solutions [18].

Using Equation (Equation 4) in (Equation 3), an ideal diffraction-free Bessel beam is obtained, i.e., a Bessel beam whose transverse intensity is independent from the propagating distance ${\left|\psi \left(\rho ,\phi ,z>0,t\right)\right|}^{2}={\left|\psi \left(\rho ,\phi ,z=0,t\right)\right|}^{2}$ [20]:(5)$$\psi \left(\rho ,z,t\right)=J_{0}\left(k_{\rho }\rho \right)e^{-jk_{z}z}e^{j\omega_{0}t}$$

As discussed in the Introduction, when generated from a finite aperture, this expression holds true only for $z\leq z_{\mathrm{ndr}}$, where $z_{\mathrm{ndr}}$ is the so-called *nondiffractive range*. According to a ray-optics analysis, this distance is given by
(6)$$z_{\mathrm{ndr}}=\rho_{\mathrm{ap}}\cot\theta_{0}$$
where $\rho_{\mathrm{ap}}$ is the aperture radius of the radiating aperture, and for $z>z_{\mathrm{ndr}}$ the field intensity abruptly decreases, i.e., $|\psi \left(\rho ,z>z_{\mathrm{ndr}},t\right){|}^{2}\to 0$. Interestingly, the transverse spot size $S_{\rho }$ of a Bessel beam is also related to the axicon angle through the following equation [21]:(7)$$S_{\rho }=0.7655\lambda \csc\theta_{0}\;$$
where λ is the vacuum wavelength. Clearly, $z_{\mathrm{ndr}}$ and $S_{\rho }$ both increase as $\theta_{0}$ decreases. In most applications, one is interested in having the largest $z_{\mathrm{ndr}}$ with the smallest $\rho_{\mathrm{ap}}$ and $S_{\rho }$, thus the choice of $\theta_{0}$ is subject to the trade-off dictated by (Equation 6) and (Equation 7) [21].

## 3. Leaky-Wave Analysis

In this section, a step-by-step design workflow is provided and justified through a simple yet rigorous theoretical explanation. In Section 3.1, a leaky-wave approach is used in order to obtain the design equations for the resonant Bessel-beam launcher presented in Section 3.2. In Section 3.3, the dispersion analysis of the device is presented.

### 3.1. The Leaky-Wave Approach

Leaky-wave antennas (LWAs) belong to the broader class of traveling-wave antennas, for which the excitation is produced by a *leaky* wave that propagates along an open waveguide-like structure [22]. The main distinctive feature of LWAs is that the radiation mechanism can efficiently be described in terms of leaky waves, which are characteristic modes endowed with complex wavenumbers, capable of providing a compact yet accurate representation of the total aperture field [22]. As shown in [5,6], a Bessel beam can be generated by means of a two-dimensional (2D) leaky radial waveguide.

Such a device is typically constituted by a grounded dielectric slab encircled by a circular metallic rim with a PRS on top, as schematically shown in Figure 1. If the PRS is but a *slight* perturbation of a perfect electric conductor (PEC), the *leaky modes* propagating in the structure will be *slight* perturbations of the guided modes existing in the unperturbed equivalent parallel-plate waveguide. Under these conditions (*small* perturbations), the total aperture field is dominated by the leaky-wave contribution, which is sufficient to provide for an accurate description of the whole radiating mechanism. (A rigorous explanation and limitations to the application of the leaky-wave approach can be found in [23,24].)

In a 2D-LWA excited by a finite three-dimensional (3D) source, the field distribution is mainly described by cylindrical leaky waves that propagate in the transverse plane with a complex radial wavenumber $k_{\rho }=\beta_{\rho }-j\alpha_{\rho }$, where $\beta_{\rho }$ and $\alpha_{\rho }$ are the phase and the leakage constant, respectively [25]. It is worthwhile recalling here that $\beta_{\rho }$ is intimately related to the axicon angle $\theta_{0}$ (which entirely defines the focusing properties of Bessel beams [21]) through the well-known ray-optics formula $\beta_{\rho }/k_{0}=\sin\theta_{0}$ [26].

As is customary, the electromagnetic problem is studied separately for TM and TE waves. In such a way, only the *z*-component of the vector potential $A_{z}$ or $F_{z}$ is required to totally describe the TM or TE field, respectively. Regardless of the polarization type, resonant Bessel-beam launchers are typically excited by a dipole-like source: if a vertical electric dipole (VED) is used a TM field distribution is achieved, whereas a TE polarization is excited through a vertical magnetic dipole (VMD).

In the case of a TM- or TE-polarized, azimuthally symmetric device, the radial dependence of $A_{z}$ or $F_{z}$, respectively, can be totally described in terms of two wave contributions: the *outward* cylindrical leaky wave excited by the dipole-like source, and the *inward* one coming from the reflection on the circular metallic rim. From an analytical viewpoint, by considering $H_{0}^{\left(1\right)}$ and $H_{0}^{\left(2\right)}$ the zeroth-order Hankel functions of the first and the second kind, respectively, it results in:(8)$$\begin{array}{cc} A_{z}\left(\rho ,z\right) & \propto A_{0}^{+}H_{0}^{\left(2\right)}\left(k_{\rho }\rho \right)+A_{0}^{-}H_{0}^{\left(1\right)}\left(k_{\rho }\rho \right) \end{array}$$(9)$$\begin{array}{cc} F_{z}\left(\rho ,z\right) & \propto F_{0}^{+}H_{0}^{\left(2\right)}\left(k_{\rho }\rho \right)+F_{0}^{-}H_{0}^{\left(1\right)}\left(k_{\rho }\rho \right) \end{array}$$
with $A_{0}^{\pm }$ ($F_{0}^{\pm }$) the complex amplitude coefficients of the outward/inward cylindrical leaky waves constituting the TM (TE) potentials. Now that we have defined the radial dependence, it remains to define the longitudinal dependence of $A_{z}$, which can simply be assumed as an outward travelling plane wave $e^{-jk_{z}z}$. The electric and magnetic field components on the aperture plane (z=0) are then found from Maxwell’s equations and, for a TM-polarized field, read:(10)$$\begin{array}{cc} \left\{\begin{array}{cc} E_{\rho }\left(\rho \right) & =-\frac{k_{z}}{\omega \epsilon }\frac{\partial }{\partial \rho }A_{z}\left(\rho \right)\\ E_{\phi }\left(\rho \right) & =0\\ E_{z}\left(\rho \right) & =-j\frac{k_{\rho }^{2}}{\omega \epsilon }A_{z}\left(\rho \right) \end{array}\right. & \;\;\;\left\{\begin{array}{cc} H_{\rho }\left(\rho \right) & =0\\ H_{\phi }\left(\rho \right) & =-\frac{1}{\rho }\frac{\partial A_{z}\left(\rho \right)}{\partial \rho }\\ H_{z} & =0 \end{array}\right. \end{array}$$

For the TE-polarized case, the duality principle can be applied, which yields null $E_{\rho }$, $E_{z}$, and $H_{\phi }$ components, with $E_{\phi }$ and $H_{\rho }$ proportional to $\frac{\partial }{\partial \rho }F_{z}$; $H_{z}$ follows the same radial envelope of $F_{z}$.

In order to obtain a Bessel beam, the aperture radius has to be properly set in such a way to have a constructive interference between the *outward* and *inward* cylindrical leaky waves that fully describe the potentials TM and TE potentials $A_{z}$ and $F_{z}$. If we assume that the coefficient of the outward wave $A_{0}^{+}$ ($F_{0}^{+}$) is almost equal to that of the inward wave $A_{0}^{-}$ ($F_{0}^{-}$), a $J_{0}\left(k_{\rho }\rho \right)$ distribution for $A_{z}$ ($F_{z}$) is achieved for TM (TE) polarization with an amplitude $A_{0}\equiv A_{0}^{+}≃A_{0}^{-}$ ($F_{0}\equiv F_{0}^{+}≃F_{0}^{-}$). As a result, the vertical field distribution $E_{z}$ ($H_{z}$) will also be characterized by a zeroth-order Bessel function. Therefore, non-null components for the TM- and TE- polarized case read:(11)$$\begin{array}{c} \left\{\begin{array}{cc} E_{z}\left(\rho ,z\right) & =-jA_{0}\frac{k_{\rho }^{2}}{\omega \epsilon }J_{0}\left(k_{\rho }\rho \right)e^{-jk_{z}z}\\ E_{\rho }\left(\rho ,z\right) & =A_{0}\frac{k_{z}k_{\rho }}{\omega \epsilon }J_{1}\left(k_{\rho }\rho \right)e^{-jk_{z}z}\\ H_{\phi }\left(\rho ,z\right) & =A_{0}\frac{k_{\rho }}{\rho }J_{1}\left(k_{\rho }\rho \right)e^{-jk_{z}z} \end{array}\right. \end{array}$$
and
(12)$$\begin{array}{c} \left\{\begin{array}{cc} H_{z}\left(\rho ,z\right) & =-jF_{0}\frac{k_{\rho }^{2}}{\omega \epsilon }J_{0}\left(k_{\rho }\rho \right)e^{-jk_{z}z}\\ H_{\rho }\left(\rho ,z\right) & =F_{0}\frac{k_{\rho }k_{z}}{\omega \mu }J_{1}\left(k_{\rho }\rho \right)e^{-jk_{z}z}\\ E_{\phi }\left(\rho ,z\right) & =-F_{0}k_{\rho }J_{1}\left(k_{\rho }\rho \right)e^{-jk_{z}z}\; \end{array}\right., \end{array}$$
respectively. Design criteria to enforce both the radial and the vertical resonance will be explained in the following subsection.

### 3.2. Design Parameters

Due to the presence of the metallic rim, a radial resonance is found when the aperture radius coincides with one of the zeroes of the Bessel functions characterizing the radial dependence of the tangential electric fields. Therefore, different resonant conditions have to be applied depending on whether we deal with TM- or TE-polarized Bessel beams. In the former case, the tangential electric-field component is $E_{z}$ that goes as $J_{0}\left(\cdot \right)$ (see (Equation 11)), whereas, in the latter case, it is $E_{\phi }$ that goes as $J_{1}\left(\cdot \right)$ (see (Equation 12)). Therefore, the aperture radius should satisfy the following equation:(13)$$\beta_{\rho }\rho_{\mathrm{ap}}=j_{nq},$$
where $j_{nq}$ is the *q*th zero of the *n*th order Bessel function, with n=0 in the TM case (as shown in, e.g., [6,10]) and n=1 in the TE case (as shown, e.g., in [13]).

From (Equation 6) and (Equation 13), a useful relation is achieved among the nondiffractive range, the aperture radius, the polarization type (which determines the order *n* of the Bessel function), and the order *q* of the radial resonance, which reads:(14)$$z_{\mathrm{ndr}}=\frac{\rho_{\mathrm{ap}}\sqrt{k_{0}^{2}\rho_{\mathrm{ap}}^{2}-j_{nq}^{2}}}{j_{nq}}\;$$

In order to obtain a Bessel-beam distribution with a desired nondiffractive range and aperture radius (fixed by practical constraints), we show here a possible design workflow to determine the required value of $\beta_{\rho }$ and $\alpha_{\rho }$ at the working frequency $f_{0}$ and the resulting design in terms of cavity height *h* and sheet reactance $X_{s}$; the latter is assumed to fully characterize a lossless PRS. This assumption is fairly accurate for canonical homogenized metasurfaces such as 2D patch arrays, metal strip gratings, and fishnet-like metasurfaces (see, e.g., [27,28]) as those discussed here. The synthesis of a fishnet-like metasurface, i.e., its practical implementation for realizing a given value of $X_{s}$, will be discussed in Section 4. Here, we first present design equations to properly set the relevant *design parameters* of a resonant Bessel-beam launcher, i.e., $X_{s}$, $\rho_{\mathrm{ap}}$, and *h*, for having a Bessel beam suitable for a given application scenario.

For this purpose, we assume the aperture radius and the nondiffractive range are fixed by some practical constraints dictated by a typical WPT scenario for wearable devices. In this context, it is commonly required to reach a minimum cover distance with a device that cannot exceed a given maximum transverse size [29], fixed in this case by the aperture radius. In this regard, Figure 2 shows through gray dashed lines how these boundaries would translate at 30, 60, and 90 GHz for a maximum aperture radius of 1 cm and a minimum cover distance of 1.5 cm. It is worthwhile to point out that, in Figure 2, the trade-off between miniaturization and cover distance for a given radial resonance is also clearly represented.

Once $f_{0}$ is fixed, the order *q* of the radial resonance and the polarization have to be chosen. In particular, the first, q=1 radial resonance is typically avoided due to its consequent high truncation effects. Equation (Equation 13) is then used to get the normalized phase constant ${\hat{\beta }}_{\rho }=\beta_{\rho }/k_{0}$, strictly related to the axicon angle through the equation ${\hat{\beta }}_{\rho }=\sin\theta_{0}$, once the polarization (n=0 or n=1 for the TM or TE case, respectively) and the radial resonance *q* are fixed.

At this point, the normalized leakage constant ${\hat{\alpha }}_{\rho }=\alpha_{\rho }/k_{0}$ has to be determined. As noted before and shown in [6], we need $A_{0}^{+}≃A_{0}^{-}\left(F_{0}^{+}≃F_{0}^{-}\right)$. Under the simplified assumption of plane-wave propagation (which ignores the radial spreading of cylindrical leaky waves) inside the cavity, the ratio $0<a_{r}<1$ between the power amplitudes of the inward and outward wave is given by $a_{r}=e^{-2\alpha_{\rho }\rho_{\mathrm{ap}}}$. In order to obtain a well-defined Bessel-beam profile without requiring the design of a very high-*Q* (*Q* stands for *quality factor* [12]) cavity, the power ratio between the waves should be moderately close to 1. On a quantitative basis, this trade-off is substantially met for $a_{r}\leq 0.95$, thus providing the following design equation for the leakage constant:(15)$${\hat{\alpha }}_{\rho }<\frac{0.03}{k_{0}\rho_{\mathrm{ap}}}$$

From the knowledge of the required ${\hat{\beta }}_{\rho }$ and ${\hat{\alpha }}_{\rho }$, the design equations for the cavity height *h* and the equivalent PRS reactance $X_{s}$ can be found for either TE- and TM- polarized Bessel-beam launchers, using one of the methods explained in [30]. For the readers’ convenience, we report here the approximate analytical equations for the reactance sheet $X_{s}$ values:
(16a)$$\begin{array}{cc} |X_{s}^{\mathrm{TE}}| & =\eta_{0}\sqrt{\frac{\pi {\hat{\beta }}_{\rho }{\hat{\alpha }}_{\rho }\sec\theta_{0}}{\sqrt{{\left(\varepsilon_{r}-\sin^{2}\theta_{0}\right)}^{3}}}} \end{array}$$
(16b)$$\begin{array}{cc} |X_{s}^{\mathrm{TM}}| & =\eta_{0}\sqrt{\frac{\pi {\hat{\beta }}_{\rho }{\hat{\alpha }}_{\rho }\cos\theta_{0}}{\varepsilon_{r}\sqrt{\varepsilon_{r}-\sin^{2}\theta_{0}}}}. \end{array}$$
and cavity height *h* values:
(17a)$$\begin{array}{cc} h_{\mathrm{TE}} & ≃\frac{\lambda_{0}}{2\sqrt{\varepsilon_{r}-\sin^{2}\theta_{0}}}\left(1-\frac{X_{s}\sqrt{\varepsilon_{r}-\sin^{2}\theta_{0}}}{\pi \eta_{0}}\right) \end{array}$$
(17b)$$\begin{array}{cc} h_{\mathrm{TM}} & ≃\frac{\lambda_{0}}{2\sqrt{\varepsilon_{r}-\sin^{2}\theta_{0}}}\left(1-\frac{X_{s}\varepsilon_{r}}{\pi \eta_{0}\sqrt{\varepsilon_{r}-\sin^{2}\theta_{0}}}\right). \end{array}$$
where $\eta_{0}≃120\pi$
Ω is the vacuum impedance. The absolute value in ([Disp-formula FD16a-micromachines-13-02230]) and () is used because the design equations apply for both inductive $X_{s}>0$ and capacitive $X_{s}<0$ sheets. We also note that Bessel-beam launchers typically work far from the leaky cutoff condition (viz., ${\hat{\beta }}_{\rho }≃{\hat{\alpha }}_{\rho }$ [31]) and usually require highly reflective PRS (which leads to small values of ${\hat{\alpha }}_{\rho }$). Therefore, the accuracy of the aforementioned equations is rather high [30]. The effectiveness of the entire design workflow can be inferred from the dispersion curves of the TE and TM leaky modes, here obtained with an accurate numerical routine, briefly discussed in the next subsection.

### 3.3. Dispersion Analysis

The propagation modes supported by the leaky-wave resonant cavity can be found by enforcing the condition of resonance on the relevant transverse equivalent network (TEN), thus obtaining the modal dispersion equation. The dispersion equation can be obtained by enforcing the resonance condition on the relevant TEN. In other words, we apply the so-called transverse resonance technique [32] to the TEN represented in Figure 3a where $Y_{0}$ and $Y_{1}$ are the characteristic admittances in air and inside the cavity, respectively, $Y_{s}=1/Z_{s}$ is the sheet admittance, and $k_{z}$ and $k_{z1}$ are the vertical wavenumber in air and inside the cavity, respectively. While the vertical wavenumbers are again given by $k_{z}=\sqrt{k_{0}^{2}-k_{\rho }^{2}}$ and $k_{z1}=\sqrt{k_{0}^{2}\varepsilon_{r}-k_{\rho }^{2}}$, the characteristic admittances have the well-known different expressions reported in Table 1 [12], depending on the specific polarization (viz., TE or TM) and material.

Therefore, we obtain the following dispersion equation:(18)$$Y_{0}\left(k_{\rho }\right)+jB_{s}-jY_{1}\left(k_{\rho }\right)\cot\left(k_{z1}\left(k_{\rho }\right)h\right)=0,$$
where $B_{s}=-1/X_{s}$, and the dependence of all terms on $k_{\rho }$ is explicit (we are tacitly assuming the PRS to be not spatially dispersive, and polarization dependent; a fair approximation for fishnet-like metasurfaces [28]). The radial wavenumbers $k_{\rho }$ of all the modes propagating in the structure are found by solving for the complex roots of (Equation 18). It should be noted that the left-hand side of (Equation 18) is a two-sheeted Riemann surface, whose complex planes are connected through the Sommerfeld branch cuts. Leaky modes are found searching for the *complex improper roots*, lying on the *improper* sheet, i.e., with Im$(k_{z})>0$. The dispersion diagrams, i.e., ${\hat{\beta }}_{\rho }$ or ${\hat{\alpha }}_{\rho }$ as functions of the frequency, are then obtained and commented for the case study analyzed in Section 4.

## 4. Structure Design

Here, we exploit the theoretical results of Section 3 to present a realistic implementation for both TE- and TM-polarized resonant Bessel-beam launchers.

The first step in the implementation of a prototype is fixing some parameters. First of all, we can assume that our device will work in a realistic scenario for WPT where it has to respect specific dimension constraints. For instance, in [33], a maximum aperture radius of 1.5 cm and a nondiffractive range of 2 cm at 30 GHz were required. In this work, a further miniaturization of the device is set by fixing $\rho_{\mathrm{ap}}=1$ cm, at the expense of a slightly lower nondiffractive range of $z_{\mathrm{ndr}}=1.5$ cm (note that the ratio $z_{\mathrm{ndr}}/\rho_{\mathrm{ap}}$ is increased). These boundaries are reported in gray dashed lines in Figure 2, where ${\bar{z}}_{\mathrm{ndr}}$ and ${\bar{\rho }}_{\mathrm{ap}}$ are the normalized nondiffractive range and aperture radius with respect to the vacuum wavelength, respectively. From Figure 2, we note that, for $f_{0}=30$ GHz, we can only use the first-order radial resonance that is preferable to avoid, as it may lead to non-negligible truncation effects. Conversely, for $f_{0}=60$ GHz and $f_{0}=90$ GHz, we can choose up to the second q=2 and third q=3 radial resonance, respectively. In particular, the latter has been considered in this work as a practical case study.

Once the radial resonance, the aperture radius, and the power ratio $a_{r}=0.95$ are fixed, the radial wavenumber $k_{\rho }$ is obtained, as previously described. With these values of $\beta_{\rho }$ and $\alpha_{\rho }$ at hand, the equivalent PRS surface impedance $Z_{s}$ and the cavity height *h* are derived through ([Disp-formula FD16a-micromachines-13-02230]) and ([Disp-formula FD16b-micromachines-13-02230]) and ([Disp-formula FD17a-micromachines-13-02230]) and ([Disp-formula FD17b-micromachines-13-02230]), respectively. All these parameters are reported for brevity in Table 2. The sought value of $Z_{s}$ is obtained through the fishnet-like unit cell shown in Figure 3b and commented in more detail in Section 4.

The theoretical effectiveness of the described design workflow is verified in Figure 3c. There, the dispersion curves of the TE and TM leaky-wave modes are reported vs. frequency *f* along with their different radial resonances. As expected, at the working frequency $f_{0}=90$ GHz, the phase-constant curves (solid lines) and the radial-resonance curves (dashed lines) intersect, thus defining the working point of the designed Bessel-beam launcher. For the TE and TM case, we found ${\hat{\beta }}_{\rho }=0.5401$ and ${\hat{\beta }}_{\rho }=0.4588$ (see Table 2), to which correspond $\theta_{0}≃32.7^{\circ }$ and $\theta_{0}≃27.3^{\circ }$ and in turn $S_{\rho }≃4.72$ mm and $S_{\rho }≃5.56$ mm, respectively.

The last steps of a the practical implementation for resonant Bessel-beam launcher are the physical realizations of a homogenized PRS with the desired equivalent $X_{s}$ and of a realistic source that acts as a VMD or a VED.

### 4.1. PRS Implementation

A PRS is a key component for different kinds of antennas: uniform 2D leaky-wave antennas, Fabry–Perot cavity antennas, reconfigurable devices and, of course, resonant Bessel-beam launchers. PRS realizations mainly fall into three different categories: single-layer dielectric covers, dielectric multilayers, and homogenized metasurfaces [34]. In order to have a low-profile antenna and to avoid high-permittivity materials that usually exhibit high dielectric losses, it is preferred to use a metal homogenized metasurface for the implementation of the PRS in Bessel-beam launchers at millimeter waves. This solution also has the advantage of being relatively low cost due to its easy realization through standard PCB technologies or, at high frequencies, standard photolithographic processes. The PRS topology, given by square or circular patches, printed dipoles, or fishnet-like metasurfaces, depends on the type of the PRS (i.e., inductive or capacitive PRS) and the desired value of the equivalent impedance sheet.

The implementation of an inductive PRS with a relatively low $X_{s}$ value can be achieved through a fishnet-like metasurface whose unit cell is reported in Figure 3b. The equivalent surface impedance can be tuned using different values for the patch gap *g* and the strip width *w*. As shown in [28], as the width increases or the gap decreases, the equivalent reactance $X_{s}$ decreases and tends to represent a PEC, as expected.

Through a parametric full-wave analysis of the unit cell, the design parameters of the fishnet-like metasurface needed to realize the desired $X_{s}$ in the TE and TM polarizations were achieved. These *g* and *w* values are reported normalized with respect to the period p=λ/10≃3.33 mm of the PRS unit cell in Table 2.

### 4.2. Feeder

So far, we have considered an ideal source given by a VED or a VMD for a TM- or a TE-polarized Bessel-beam launcher, respectively. This kind of source, however, is not a realistic feeder, thus it is necessary to take into account its physical implementation.

As concerns the VED, a low-cost, effective, simple solution is given by coaxial probes. In fact, coaxial probes behave as VED-like sources and are commercially available up to 110 GHz. Conversely, realizations of VMDs present different issues. In [13,14], loop antennas or coils were considered for exciting a TE-polarized Bessel beam. However, the feeding point of this kind of source necessarily breaks the azimuthal symmetry of the structure, deteriorating the TE polarization purity. The solution proposed in [13] that partially solves this issue is given by a specific shift of the probe position from the center obtained through a full-wave parametric optimization. In such a way, a zeroth-order Bessel-beam profile is achieved for the vertical component of the magnetic field, but spurious contributions are still present. It was recently shown [17] that it is possible to excite a purely TE-polarized Bessel-beam launcher by means of a radial, simultaneously excited slot array on the ground plane. Each slot can be excited by a microstrip, but this may need a rather complex feeding network, because the number of ports increases with the number of slots.

An innovative single-port excitation for a TE-polarized Bessel-beam launcher is then presented here. Whereas a radial slot array is the discrete counterpart of a radially directed magnetic surface current, we can excite this source in its continuous form by means of a ${\mathrm{TE}}_{01}$-polarized circular waveguide. The ${\mathrm{TE}}_{01}$ mode is a higher-order mode in a circular waveguide, and it can be isolated through different implementations described in the available literature [35,36,37]. In order to reduce the computational burden, we have considered on CST Microwave Studio [38] an ideal waveguide port where only the ${\mathrm{TE}}_{01}$ mode is excited by ensuring, through the guide dimension, that the desired mode is above cutoff.

### 4.3. Comparison between Theoretical and Full-Wave Results

The consistency of the proposed approach is verified through a comparison between full-wave results and the numerical evaluation of expected theoretical field distributions according to the outlined leaky-wave approach. The former are obtained through a CST simulation of the entire three-dimensional model of the structure for both polarizations; the latter are obtained through the application of a Gauss–Legendre quadrature rule to the radiation integral of the theoretical aperture fields.

In particular, for the TM-polarized case, the aperture field distribution is given by (Equation 11) in z=0. From the knowledge of the field, the equivalent magnetic and electric surface currents over the aperture plane are derived and used in the radiation integral to get the near-field distribution for z≥0 [39]. The typical profile of a zeroth-order Bessel beam is obtained for the longitudinal component of the electric field $E_{z}$, as shown in Figure 4, where the field amplitude over the ρz-plane is reported in dB, normalized with respect to its maximum, for both numerical and full-wave results. In both cases, a Bessel beam with a transverse spot size of about half a centimeter is maintained over a distance of 1.5 cm. An impressive agreement between theoretical and full-wave results is obtained, thus corroborating the effectiveness of the presented design workflow. In Figure 4, we can appreciate the typical on-axis oscillations of truncated Bessel beams. To circumvent this issue, and thus generate a quasi-homogeneous axial intensity distribution, apodization techniques (see, e.g., [40,41]) are often proposed in optics, as well as at microwave frequencies (see, e.g., [42,43]).

As a consequence of the duality principle, similar results are obtained for the TE-polarized case shown in Figure 5, where the longitudinal component of the magnetic field $|H_{z}|$ is represented as per $|E_{z}|$ in Figure 4. In this case, the theoretical approach is the same used for the TM case, provided the aperture field distribution is now given by (Equation 12) [39].

## 5. Applications

In the previous sections, the theoretical implementation and the design workflow of resonant Bessel-beam launchers based on the leaky-wave approach were described and validated through full-wave simulations. Here, we show different practical scenarios that may take great advantage of such launchers.

Bessel beams, due to their remarkable features of beam focusing and self-healing in the near field, are very useful in diverse engineering fields [44]. For instance, high data-rate links or micromanipulation of small particles can be achieved through Bessel beams, as shown in [45,46], respectively. Bessel beams are also used in tomography, imaging [47,48,49], and ground penetrating radar applications [50,51], also with interesting reconfigurable properties [52]. Moreover, Bessel beams have attracted much interest in the THz range [53] for improving the depth-of-field [54]. They can be implemented through microstructured photoconductive antennas [55], plasmonic waves [56], or irregular binary axicons [57]. Interestingly, in the 3–300 GHz frequency range, Bessel-beam launchers present a very low-cost and simple implementation and a higher penetration depth with respect to their optical counterparts.

In this regard, WPT is one of the application scenarios where the generation of Bessel beams at 3–300 GHz has attracted much interest in the last decade. In particular, the Bessel-beam limited-diffractive behavior can be exploited in order to realize a radiative near-field WPT link. The radiative near-field link presents a higher efficiency with respect to far-field WPT links [58] and a higher covered distance with respect to the typical reactive near-field links based on inductive coupling [59]. As explained in [60], for distances beyond the reactive near field, magneto-inductive coupling becomes insignificant due to the decay of the link efficiency as the sixth power of the distance between transmitter and receiver.

Bessel beams have recently been studied for the implementation of radiative near-field WPT links [16,37]. In practical cases, such as the realization of wearable devices, it is very important to consider a compact structure for realizing a WPT link over a few wavelengths distance. For instance, in [17,29,33], a miniaturized radiative near-field link was established between two resonant Bessel-beam launchers. In such a way, a direct link between these two resonant devices was achieved, such as the one represented in Figure 6.

By exploiting the TM-polarized Bessel-beam design presented in this work, an innovative WPT link at 90 GHz can be realized. As shown in Figure 6a, a near-field radiative WPT link is realized between two identically polarized resonant Bessel-beam launchers placed one in front of the other. By considering different distances between such devices and assuming the link as a two-port network, the resulting $|S_{21}|$ is reported in Figure 6b. In particular, the link has been simulated on CST Microwave Studio at 90 GHz for different distances between the devices (5, 10, 15, and 20 mm), obtaining interesting and promising results for high-frequencies WPT links.

Moreover, Bessel beams can show very different features depending on the specific application. If the cover distance and the size of the launcher are the main concerns, Bessel beams can be realized to work with a very low axicon angle. This, however, comes at the expense of the transverse size of the beam waist, which increases for lower axicon angles [21]. Conversely, one is keen to use a higher axicon angle when the transverse resolution is the main concern and the cover distance is not very important. On the other hand, if it is required to maintain a narrow beam waist for large distances, the size of the launcher has to be increased. In such conditions, resonant Bessel-beam launchers are not the best choice, and it is preferred to resort to the wideband type (see, e.g., [7,61,62,63]) mentioned in the Introduction.

In any case, the limited-diffraction and self-healing character of Bessel beams will always represent an added value with respect to more common realizations for applications where it is important to avoid waste of power and to ensure coverage in non-line-of-sight scenarios.

## 6. Conclusions

In this work, we started from a very general theoretical framework to analyze Bessel beams as ideal nondiffractive solutions to the wave equation. Their practical generation at millimeter waves through planar devices commonly known as resonant Bessel-beam launchers was presented and explained under the frame of a rigorous leaky-wave analysis. The latter has allowed for deriving simple, yet accurate analytical equations for developing a useful design workflow for such devices.

The effectiveness of such design guidelines were corroborated through full-wave simulations of a device of only 2 cm of transverse size capable of generating Bessel beams at 90 GHz that maintain a transverse spot size less than half a centimeter over a 1.5 cm distance.

The potentialities of the proposed device were finally tested in a realistic scenario, where two identical resonant launchers were placed one in front of the other to create an innovative wireless link at 90 GHz in the radiative near-field. The performance of such a link opens interesting perspectives for future applications of these devices in wireless-power-transfer and other focusing applications. 

## Figures and Tables

**Figure 1 micromachines-13-02230-f001:**
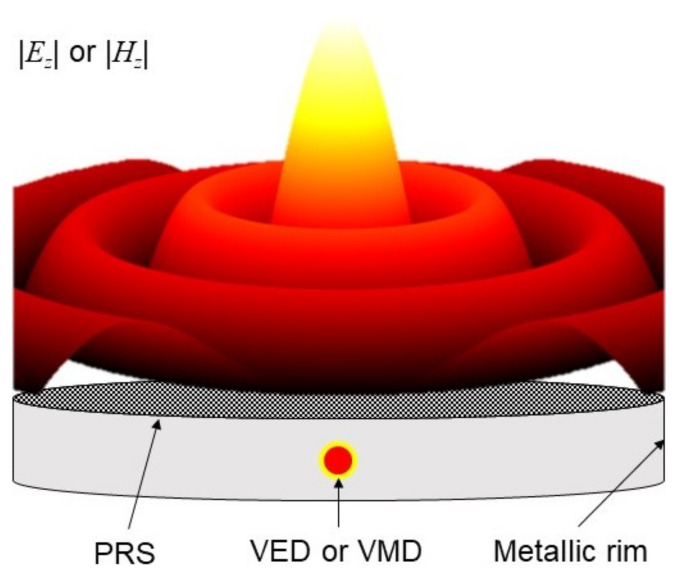
Pictorial representation of a TM- or TE-polarized resonant Bessel-beam launcher excited by a VED or a VMD with its $|E_{z}|$ or $|H_{z}|$ near-field distribution, respectively.

**Figure 2 micromachines-13-02230-f002:**
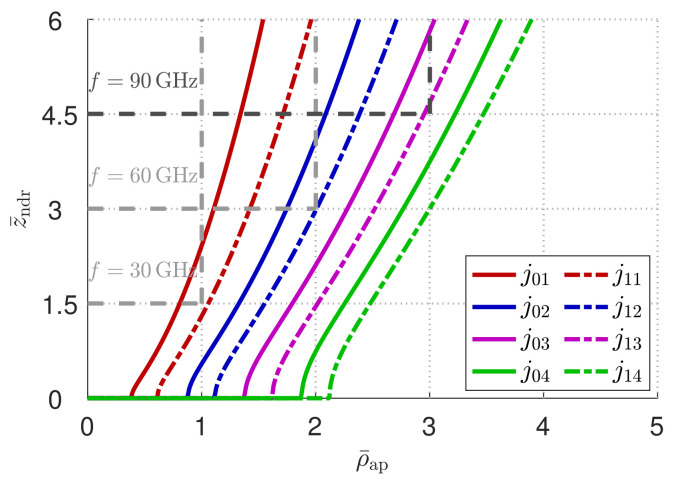
Design curves for a minimum $z_{\mathrm{ndr}}=15$ mm and a maximum $\rho_{\mathrm{ap}}=10$ mm. The normalized nondiffractive range ${\bar{z}}_{\mathrm{ndr}}=z_{\mathrm{ndr}}/\lambda_{0}$ and the normalized aperture radius ${\bar{\rho }}_{\mathrm{ap}}=\rho_{\mathrm{ap}}/\lambda_{0}$ are reported on the *y*-axis and the *x*-axis, respectively. Gray dashed lines represent the design boundaries at 30, 60, and 90 GHz, and differently colorized solid or dashed lines represent the design curves in a TM or TE polarization, respectively, for different radial resonances.

**Figure 3 micromachines-13-02230-f003:**
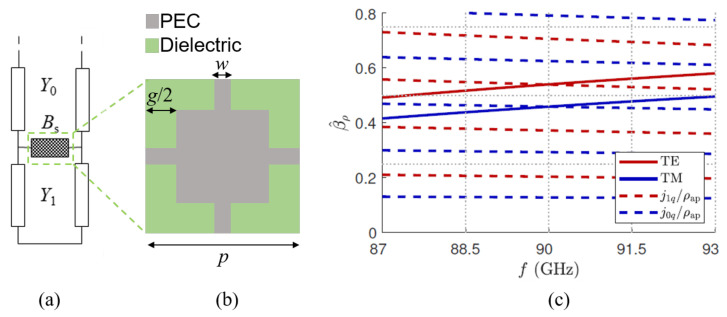
(**a**) Transverse equivalent network of a resonant Bessel-beam launcher, and (**b**) the geometry of the PRS fishnet-like unit cell. The areas in which PEC is not present are filled by the dielectric inside the cavity, constituted by air in this case. (**c**) Dispersion diagrams of the normalized phase constant ${\hat{\beta }}_{\rho }$ vs. frequency *f* for the specific case derived in Section 4, with the design parameters reported in Table 2. Red and blue solid lines represent the dispersion curves of TE and TM leaky modes, respectively. The radial resonances for TE and TM modes, represented by red and blue dashed lines, are obtained by enforcing a null of the first- or the zeroth-order Bessel function, respectively.

**Figure 4 micromachines-13-02230-f004:**
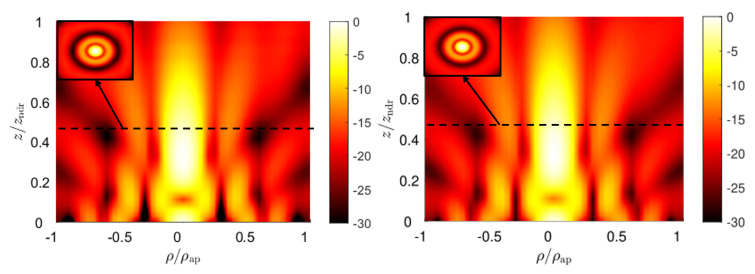
Full-wave and radiation-integral results for the longitudinal electric-field component $E_{z}$ in the TM-polarized Bessel-beam launcher on the left and on the right, respectively. At the top-left corner, the $|E_{z}|$ field distribution is represented over the transverse xy-plane at $z=z_{\mathrm{ndr}}/2$.

**Figure 5 micromachines-13-02230-f005:**
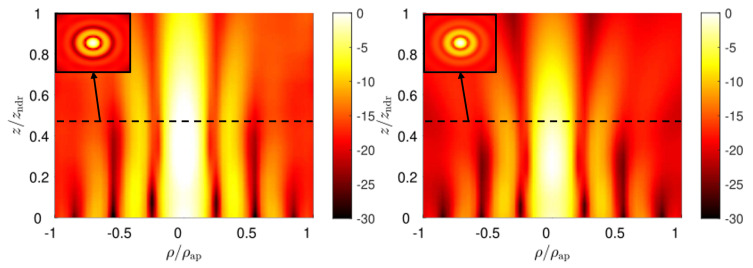
Full-wave and radiation-integral results for the longitudinal magnetic-field component $H_{z}$ in the designed TE-polarized Bessel-beam launcher on the left and on the right, respectively. At the top-left corner, the $|H_{z}|$ field distribution is represented over the transverse xy-plane at $z=z_{\mathrm{ndr}}/2$.

**Figure 6 micromachines-13-02230-f006:**
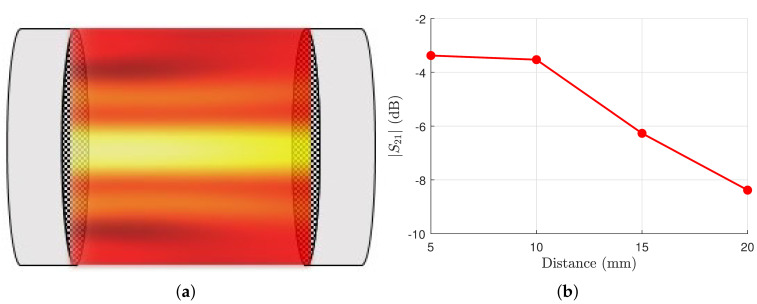
(**a**) Pictorial representation of a radiative near-field WPT link between two resonant Bessel-beam launchers. (**b**) The $|S_{21}|$ values in a radiative near-field WPT link at 90 GHz for different distances between transmitting and receiving TM-polarized Bessel-beam launchers.

**Table 1 micromachines-13-02230-t001:** Expression for the characteristic admittances in different material and polarizations.

Admittance	TE	TM
$Y_{0}$	$\frac{k_{z}}{k_{0}\eta_{0}}$	$\frac{k_{0}}{k_{z}\eta_{0}}$
$Y_{1}$	$\frac{k_{z1}}{k_{0}\eta_{0}}$	$\frac{k_{0}\varepsilon_{r}}{k_{z1}\eta_{0}}$

**Table 2 micromachines-13-02230-t002:** Design parameters and radial wavenumbers for TE- and TM-polarized resonant Bessel-beam launchers with a working frequency $f_{0}=90$ GHz and an aperture radius $\rho_{\mathrm{ap}}=10$ mm. The period of the PRS unit cell is p≃3.33 mm.

Polarization	${\hat{\beta }}_{\rho }$	${\hat{\alpha }}_{\rho }$	$X_{s}$ (Ω)	*h* (mm)	g/p	w/p
TE	0.5401	0.0019	30	1.94	0.2086	0.05
TM	0.4588	0.0020	20	1.84	0.1732	0.1

## Data Availability

Not applicable.

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
