# Peer review of "A Leaky-Wave Analysis of Resonant Bessel-Beam Launchers: Design Criteria, Practical Examples, and Potential Applicationsat Microwave and Millimeter-Wave Frequencies"

_micromachines, 2022, doi:10.3390/mi13122230_

Round 1

Reviewer 1 Report

In this paper,  the work done by the authors is relevant and a lot of fruitful work has been done in terms of research. The structure of the paper is reasonable, and the research methods are appropriate. But I think this paper needs to be improved as follows:

1.      Some of the references in this paper are rather old and need to be cited more recently.

2.      The authors should explain in detail why only the resonant kind is discussed and not the wideband kind.

3.      The authors should clearly present the highlights of this paper in the introduction section.

Reviewer 2 Report

Notes are in the file.

Round 2

Reviewer 2 Report

Thanks to the authors for the complete and correct answers to the questions that have arisen. I believe that the article can be published in the journal Micromachines. I wish the authors not to stop and continue their research.